# LOTION: SMOOTHING THE OPTIMIZATION LANDSCAPE FOR QUANTIZED TRAINING

## ABSTRACT

Optimizing neural networks for quantized objectives is fundamentally challenging because the quantizer is piece-wise constant, yielding zero gradients everywhere except at quantization thresholds where the derivative is undefined. Most existing methods deal with this issue by relaxing gradient computations with techniques like Straight Through Estimators (STE) and do not provide any guarantees of convergence. In this work, taking inspiration from Nesterov smoothing, we approximate the quantized loss surface with a continuous loss surface. In particular, we introduce LOTION, **L**ow-precision **O**ptimization via s**T**ochastic-no**I**se sm**O**othi**N**g, a principled smoothing framework that replaces the raw quantized loss with its expectation under unbiased randomized-rounding noise. In this framework, standard optimizers are guaranteed to converge to a local minimum of the loss surface. Moreover, when using noise derived from stochastic rounding, we show that the global minima of the original quantized loss are preserved. We empirically demonstrate that this method outperforms standard QAT on synthetic testbeds and on 150M- and 300M- parameter language models.

## 1 INTRODUCTION

Although the performance of Large Language Models scales predictably with the size of the model (Hoffmann et al., 2022; Kaplan et al., 2020), these gains come with a corresponding cost when the model is deployed for inference (Sardana et al., 2025; Zhu et al., 2024). During deployment, the primary bottleneck is streaming billions of parameters along the memory hierarchy. Because decoding latency is dominated by this memory traffic, inference is overwhelmingly bandwidth-bound (Austin et al., 2025). As a result, model compression and low-precision computation are becoming the default for training and serving LLMs on modern accelerators (Kumar et al., 2024; Micikevicius et al., 2022).

Post-training quantization (PTQ) and quantization-aware training (QAT) directly alleviate this burden by compressing model weights and/or activations to low-precision formats while optimizing for quantized performance. However, when a quantizer is naively inserted into the optimization loop, the training objective becomes highly discontinuous: every forward pass maps weights according to a finite codebook, zeroing gradients almost everywhere. PTQ remedies this issue by training in full precision and optimizing simpler surrogate objectives (e.g. layer- or matrix-multiply-specific reconstruction), but it usually underperforms QAT, particularly at lower bit-widths. QAT instead optimizes the quantized objective by relying on the straight-through estimator (STE), which treats the non-differentiable quantizer as the identity in the backward pass.

Despite some empirical successes, naive identity-based STEs provide no theoretical guarantees and tend to become unstable in recent low-precision formats that quantize more aggressively, motivating a more principled alternative (Spallanzani et al., 2022; Gong et al., 2019; Liu et al., 2018; Nagel et al., 2022; Park & Yoo, 2020; Sakr et al., 2022; Shin et al., 2023; Yin et al., 2019). We propose a fully principled, parameter-free alternative that is applicable to a wide variety of rounding schemes.

We propose **LOTION**—Low-precision **O**ptimization via s**T**ochastic-no**I**se sm**O**othi**N**g. Instead of modifying the gradient, LOTION directly smooths the *loss* itself: it trains on the expectation of the quantized loss under randomized-rounding noise. This expectation is differentiable almost

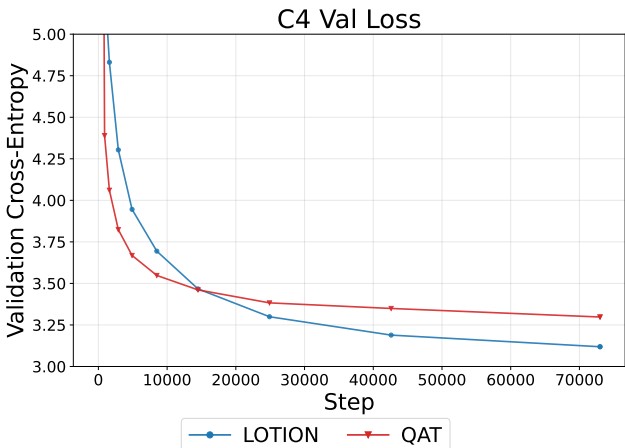

Figure 1: Quantized validation loss at INT4 precision for LOTION and QAT on a 150M-parameter model. We quantize checkpoints with round-to-nearest (RTN) and randomized rounding (RR) and, for each method, plot the variant that yields the lowest validation loss. The full plot can be found in Figure 10.

everywhere, so any standard first- or second-order optimizer can be used with its usual convergence guarantees (in particular, non-smooth convergence guarantees (Davis et al., 2018)).

**Our contributions.** The main contributions of this work are as follows:

- We introduce **LOTION**, a loss-smoothing framework that replaces discontinuous quantized objectives with an almost everywhere differentiable surrogate obtained via randomized rounding. Additionally, we show that this framework preserves *all* global minima of the original quantized problem.

- For neural networks, we derive a closed form expression for the second-order approximation of the obtained loss. We show that this approximation can be interpreted as a curvature-aware regularizer, providing a transparent interpretation of how randomized rounding stabilizes quantized training.

- As a proof of concept, we show that LOTION significantly improves upon the accuracy of widely used STE-based QAT and PTQ methods at INT4 precision on a synthetic linear regression task.

- Finally, we pre-train 150M- and 300M- parameter language models and quantize them to INT4, INT8, and FP4. We include our 150M parameter experiment in Figure 1, showing that LOTION achieves lower post-quantization validation loss than PTQ and QAT baselines.

## 1.1 RELATED WORK

**Quantization-Aware Training (QAT) vs Post-Training Quantization (PTQ).** Neural network quantization is typically performed either during training (QAT) or after training (PTQ). QAT methods often rely on the *straight-through estimator* (STE)–a heuristic gradient approximation formalized by Bengio et al. (Bengio et al., 2013). STE enables gradient-based training by pretending that the quantizer is the identity during backpropagation. Despite its popularity, the STE lacks convergence guarantees and is known to cause gradient instability, especially at 4- or 2-bit precision (Huh et al., 2023; Yin et al., 2019). Numerous works attempt to mitigate this via gradient scaling, learned quantization scales (Esser et al., 2020), or progressive bit-reduction schedules (Nagel et al., 2020). PTQ methods, by contrast, quantize pretrained models post hoc and optimize auxiliary calibration objectives. Layer-wise curvature metrics, such as the Hessian trace (Dong et al., 2019), or sensitivity analyses are often used to allocate bits across layers. While PTQ avoids instability, it often underperforms QAT at very low precision.

**Loss Smoothing via Noise and Stochastic Rounding.** Another line of work introduces noise to smooth the quantized objective, sidestepping non-differentiability. This idea is rooted in classical techniques such as Nesterov smoothing (Nesterov, 2005) and randomized smoothing (Duchi et al., 2012; 2011). In neural network quantization, additive noise has been used to simulate quantization effects during training (Baskin et al.; Défossez et al., 2022), with improved empirical stability. Recent methods extend this idea to fully differentiable quantization-aware training. For instance, NIPQ (Shin et al., 2023) replaces the STE with a noise-based proxy and learns both the scales and bit-widths. While this proxy enables end-to-end training, it introduces additional parameters that require additional tuning. Furthermore, NIPQ samples noise from the quantization-error distribution (or, in practice, uniform noise), which only approximates the true parameter-dependent error and does not capture its full structure. These noise proxies lack convergence guarantees and typically require manual tuning.

**Quantization and Curvature-Awareness.** Several works relate quantization error to curvature of the loss landscape. For instance, HAWQ (Dong et al., 2019) and its successors use the Hessian spectrum to guide bit allocation. Other approaches estimate layer sensitivity using Hutchinson-based Hessian trace approximations (Zhou et al., 2018). Some training-time regularization methods penalize sharp minima (Nagel et al., 2022), promoting flatness to better tolerate quantization. These methods highlight the importance of curvature in quantization-aware optimization, but often rely on global heuristics or require non-standard solvers.

**Our Approach: LOTION.** LOTION introduces a loss-level smoothing framework that avoids heuristic gradient modifications. Instead of backpropagating through a non-differentiable cast function, LOTION directly optimizes the *expected quantized loss* under unbiased randomized rounding noise. This yields a continuous, almost-everywhere differentiable objective that preserves all global minima of the original quantized loss and supports a broad range of rounding schemes. For neural networks, we provide a closed form expression for the second-order approximation of LOTION, making explicit the connection between quantization noise and second-order structure. This principled formulation provides both empirical stability and theoretical guarantees absent from prior methods.

## 2 PROBLEM SETTING

We adopt the standard supervised learning setup. Let $(x, y) \sim \mathcal{D}$ denote input–label pairs and let $f(w; x)$ be the output of a neural network parameterized by weights $w \in \mathbb{R}^d$. With a per-example loss $\ell(\cdot, \cdot)$, the population loss is

$$\mathcal{L}(w) = \mathbb{E}_{(x,y) \sim \mathcal{D}} \big[ \ell\big(f(w; x), y\big) \big].$$

Quantization constrains us to a finite codebook of representable weights. Let $\mathrm{cast} : \mathbb{R}^d \to Q$ denote the quantization operator mapping real-valued weights to $Q \subset \mathbb{R}^d$. We therefore seek to minimize the loss on the quantized model:

$$\min_{w \in \mathbb{R}^d} \mathcal{L}\big(\mathrm{cast}(w)\big).$$

**Remark 1.** *The mapping $w \mapsto \mathcal{L}\big(\mathrm{cast}(w)\big)$ is piecewise constant; gradients vanish except on the measure-zero cell boundaries induced by the quantizer, where the gradients do not exist.*

### 2.1 FINE-GRAINED SHARED-SCALE INTEGER QUANTIZATION

We study the standard symmetric signed integer quantization method in which parameters are scaled into the desired dynamic range and partitioned into uniform blocks as in the absolute maximum quantization method described in LLM.INT8() (Dettmers et al., 2022). We prefer absolute maximum quantization because it prevents overflow and avoids the computation and memory burden of methods like zero-point asymmetric schemes that are particularly costly for weight-only quantization. Absolute max quantization is often used in practice, notably in the FP8 "fine-grained shared-scale" format adopted by DEEPSEEK (DeepSeek-AI et al., 2025). Parameters are partitioned into blocks $B$ (possibly as small as a single element). For each input block $w_i$ we store one floating-point (FP16) scale $s_B$ and an $n$-bit integer tensor $z$ defined by

$$s_B = \frac{\max_{i \in B} |w_i|}{2^{n-1} - 1}, \qquad z_i = \left\lfloor \frac{w_i}{s_B} \right\rceil, \qquad [\mathrm{cast}(w)]_i = s_B z_i \quad (i \in B).$$

Because $|w_i| \leq (2^{n-1}-1)\, s_B$ by construction, the rounded values satisfy $z_i \in [-(2^{n-1}-1), 2^{n-1}-1]$ and thus lie safely within the representable range; no explicit clipping step is required.

The scale parameters $\{s_B\}$ depend on $w_i$, so the quantization lattice moves as optimization proceeds. This coupling between the geometry of $Q$ and the training dynamics is central to the loss stability analyzed later. We choose to keep our scale parameters in high precision, but we can easily extend LOTION to other quantization formats where the scale parameters are differentiable with respect to the weights.

## 3 LOTION: SMOOTHING THE LOSS

The core idea of our smoothing approach is to turn our non-differentiable (and discontinuous) optimization problem into a continuous one. In particular, the approach is to consider a stochastically perturbed optimization problem of the form:

$$\mathcal{L}_{\text{smooth},D}(w) = \mathbb{E}_{q \sim D_w}[\mathcal{L}(q)]$$

where $D_w$ represents a distribution over the points in $Q$, where the distribution is allowed to depend on $w$. For example, the Gaussian smoothing approach (analyzed by Nesterov (2005)) would be to first sample $\varepsilon \sim N(0, \sigma^2 I)$, and then take $q = \text{cast}(w + \varepsilon)$. For this choice of $D_w$, then $\mathcal{L}_{\text{smooth},D}$ will be a continuous and differentiable function for all orders.

In this work, we will consider a randomized rounding approach, formally defined in Section 3.1, which lets us make connections to prior work and helps us derive a more principled regularization approach.

### 3.1 RANDOMIZED ROUNDING

To define a general notion of randomized rounding, let us first denote the support of the cast function as $Q$, i.e, $Q = \{x \mid \exists w \text{ s.t. } \text{cast}(w) = x\}$. Also, let us define a notion of distance between two finite sets $(S_1, S_2)$ of points in $\mathbb{R}^d$ as $d(S_1, S_2) = \max\{\|w_1 - w_2\|_2 \mid w_1 \in S_1, w_2 \in S_2\}$.

**Definition 1** (Randomized Rounding). *Randomized Rounding (RR) is a function from $\mathbb{R}^d \to \mathbb{P}[Q]$ that satisfies the following three properties:*

1. $\forall w \in \mathbb{R}^d, \mathbb{E}_{q \sim \text{RR}(w)}[q] = w$

2. *RR is continuous and locally bounded [1], where the continuity is defined with respect to $\mathcal{W}_2$ distance on $\mathbb{P}[Q]$ and $L_2$ distance on $\mathbb{R}^d$.*

3. $\forall w \in Q$ *which satisfy $cast(w) = w$, $\text{RR}(w)$ has probability $1$ on $w$.*

The smoothed loss defined with respect to randomized rounding satisfies some nice properties.

**Lemma 1.** *For any loss function $L(w)$ which is continuous w.r.t $L_2$ norm and any $f : \mathbb{R}^d \to \mathbb{P}[Q]$ satisfying the 2nd axiom above, $\mathbb{E}_{q \sim f(w)}[L(q)]$ is also continuous w.r.t the $L_2$ norm.*

**Lemma 2.** *For any $f : \mathbb{R}^d \to \mathbb{P}[Q]$ which satisfies the 3rd axiom above,*

$$min_{w \in \mathbb{R}^d} \mathbb{E}_{q \sim f(w)}[L(q)] = min_{w \in \mathbb{R}^d} L(cast(w))$$

The proofs for Lemma 1 and Lemma 2 are provided in Appendix A.2. The above two lemmas combined show that $\mathbb{E}_{q \sim \text{RR}(w)}[L(q)]$ is a continuous function whose global minima match the global minima of the quantized loss function. Thus, we have a better, optimizable function that does not affect the global minimum of the loss surface.

### 3.2 WARM-UP: QUADRATIC LOSSES

To visualize how we incorporate randomized rounding into LOTION, we first consider a setting where the (population) loss is quadratic

$$\mathcal{L}(w) \;=\; \tfrac{1}{2}\,(w - w^\star)^\top H\,(w - w^\star), \qquad H \succeq 0.$$

---

[1]Locally bounded means that for any compact set $D \subset \mathbb{R}^d$, $d(\text{spt}(\text{RR}(D)), \{0\})$ is finite

Let $\varepsilon$ denote the *randomized-rounding noise*, so $\mathrm{RR}(w) = w + \varepsilon$ with $\mathbb{E}[\varepsilon] = 0$ and $\Sigma_\varepsilon := \mathrm{Cov}[\varepsilon]$. Because the randomized rounding noise is zero-mean, expanding the quadratic and taking expectations gives a closed form:

$$\boxed{\mathcal{L}_{\mathrm{smooth}}(w) \;=\; \mathbb{E}_\varepsilon\big[\mathcal{L}(w + \varepsilon)\big] \;=\; \mathcal{L}(w) + \tfrac{1}{2}\,\mathrm{tr}\big(H\,\Sigma_\varepsilon\big)} \tag{1}$$

**Covariance of unbiased randomized rounding.**   For the fine-grained shared-scale rule in Section 2.1, each coordinate $i$ (belonging to block $B(i)$ with scale $s_B$) is rounded independently:

$$z'_i \;=\; \frac{w_i}{s_B}, \quad \varepsilon_i \;=\; s_B\big(R_i - z'_i\big), \quad R_i = \begin{cases} \lfloor z'_i \rfloor & \text{w.p. } \lceil z'_i \rceil - z'_i, \\ \lceil z'_i \rceil & \text{w.p. } z'_i - \lfloor z'_i \rfloor. \end{cases}$$

Writing $\Delta_i := z'_i - \lfloor z'_i \rfloor \in [0, 1]$,

$$\mathrm{Var}[\varepsilon_i] \;=\; s_B^2\, \Delta_i\,(1 - \Delta_i).$$

Since distinct coordinates round independently, $\Sigma_\varepsilon = \mathrm{diag}(\sigma_1^2, \ldots, \sigma_d^2)$ with $\sigma_i^2 = s_{B(i)}^2 \Delta_i (1 - \Delta_i)$.

**Interpretation: An Implied Regularizer**   Plugging this diagonal covariance into equation 1 yields an $\ell_2$-style *ridge* term:

$$\mathcal{L}_{\mathrm{smooth}}(w) \;=\; \mathcal{L}(w) \;+\; \tfrac{1}{2}\sum_{i=1}^{d} H_{ii}\, s_{B(i)}^2\, \Delta_i(1 - \Delta_i).$$

Thus, randomized rounding *exactly* adds a data-dependent diagonal regularizer whose strength is dependent on the curvature of the hessian and the expected rounding error in the given coordinate. This makes the new objective strictly smoother than the original, yet preserves all global minima (See Lemma 1 and 2).

We also analyze an QAT-like algorithm for the quadratic loss. In QAT, the gradient is computed on a quantized point and used to update the full-precision parameters. We define an analogous algorithm RAT (Rounding-Aware Training) that computes gradients on *randomly rounded* points instead.

**Lemma 3.** *For $\varepsilon$ drawn from randomized rounding noise and quadratic loss,*

$$\mathbb{E}_\varepsilon[\nabla\mathcal{L}(w + \varepsilon)] = \nabla\mathcal{L}(w)$$

The above lemma shows that RAT with a quadratic loss simply provides an unbiased estimate of the original gradient, implying that RAT would perform similarly to PTQ in this setting. By extension, this shows that methods like QAT are also likely to perform comparably with PTQ at best for quadratic losses and seem to be less principled than LOTION.

## 3.3   General Case and a Gauss–Newton Regularizer

Building on the quadratic analysis, we define the smoothed objective

$$\mathcal{L}_{\mathrm{smooth}}(w) \;:=\; \mathbb{E}_{\varepsilon\sim\mathrm{RR}(w)}\big[\mathcal{L}(w + \varepsilon)\big],$$

where $\mathrm{RR}(w)$ is the unbiased randomized-rounding distribution formally defined in Section 3.1. Under random rounding, a parameter is rounded up or down with probability corresponding to the distance from the upper and lower quantization bin. For a twice-differentiable loss we have the second-order expansion

$$\mathcal{L}(w + \varepsilon) = \mathcal{L}(w) + g(w)^\top \varepsilon + \tfrac{1}{2}\,\varepsilon^\top H(w)\varepsilon + \mathcal{O}\big(\|\varepsilon\|^3\big),$$

with gradient $g(w) = \nabla\mathcal{L}(w)$ and Hessian $H(w) = \nabla^2\mathcal{L}(w)$. Taking expectations and using $\mathbb{E}[\varepsilon] = 0$ yields

$$\mathcal{L}_{\mathrm{smooth}}(w) = \mathcal{L}(w) + \tfrac{1}{2}\,\mathrm{tr}\big(H(w)\,\Sigma_\varepsilon(w)\big) + \mathcal{O}\big(\mathbb{E}[\|\varepsilon\|^3]\big),$$

where $\Sigma_\varepsilon(w) = \mathrm{Cov}[\varepsilon]$.

**Gauss–Newton replacement.** The Hessian of the neural network $f$ can be decomposed as a sum of two terms:

$$\nabla_w^2 \ell = \underbrace{\nabla_w f^T \nabla_f^2 \ell \nabla_w f}_{G(w)} + \nabla_w \ell \nabla_w^2 f$$

where the first component is positive semi-definite for convex losses and is referred to as the Gauss-Newton component. As the full Hessian may introduce negative curvature, we therefore substitute it with the positive-semidefinite *Gauss–Newton* matrix. Dropping higher-order terms gives the working approximation

$$\mathcal{L}_{\mathrm{GN}}(w) = \mathcal{L}(w) + \tfrac{1}{2} \operatorname{tr}\big(G(w)\,\Sigma_\varepsilon(w)\big) \tag{2}$$

**Diagonal form under unbiased rounding.** The random rounding scheme is coordinate-wise, so $\Sigma_\varepsilon(w) = \operatorname{diag}(\sigma_1^2, \ldots, \sigma_d^2)$ with

$$\sigma_i^2 = s_{B(i)}^2 \Delta_i (1 - \Delta_i),$$

where $s_{B(i)}$ is the shared scale of the block $B(i)$ being rounded and $\Delta_i \in [0,1]$ is the fractional part of $w_i / s_{B(i)}$, the distance to the lower quantization bin after scaling (see Section 3.1 for more details). Writing $g_{ii}(w)$ for the $i$th diagonal element of $G(w)$,

$$\mathcal{L}_{\mathrm{GN}}(w) = \mathcal{L}(w) + \frac{1}{2} \sum_{i=1}^d g_{ii}(w)\, \sigma_i^2 = \mathcal{L}(w) + \frac{1}{2} \sum_{i=1}^d g_{ii}(w)\, s_{B(i)}^2 \Delta_i (1 - \Delta_i). \tag{3}$$

**Interpretation and optimization.** Equation 3 shows that randomized rounding injects an $\ell_2$-style *curvature-aware* ridge regularizer. Thus, randomized rounding penalizes quantization in sharp directions. In practice, the diagonal terms of the Gauss-Newton component can be obtained by either using another backpropagation with sampled labels as done in Sophia (Liu et al., 2024) or we can use the empirical Fisher approximation by accumulating the square of the gradients observed in practice as done by Adam (Kingma & Ba, 2017).

## 4 EXPERIMENTS

In this section, we provide both synthetic and large language model experiments to compare the performance of our smoothed loss with analogous QAT and PTQ baselines. The synthetic experiments are lightweight and were run on A100s/H100s in less than an hour per run. Both the synthetic and language model experiments can run on any modern hardware, as all computations are done in FP32 with simulated weight quantization or rounding. The PTQ runs are trained end-to-end in FP32, and model checkpoints are quantized (standard round-to-nearest quantization, see Section 2.1) or rounded (randomized rounding, see Section 3.1) for evaluations. The QAT baseline simulates standard weight quantization and Rounding Aware Training (RAT) simulates randomized rounding in the forward pass. Both perform backward pass operations in full precision using the straight-through estimator. We train LOTION in full precision and round for evaluations. All hyperparameters tested are described in the Appendix A.5. Given a tensor of weights and a precision format, we scale the entire tensor into the dynamic range of the format by using the standard absmax scaling factor described in Section 2.1. Then for each parameter in the tensor, we calculate $\operatorname{Var}[\varepsilon_i] = s_B^2 \Delta_i (1 - \Delta_i)$ by using the per-tensor shared scaling factor and the distance of the resulting scaled parameter from the nearest two quantization bin boundaries. As the unbiased randomized rounding is applied to each parameter independently, this can be done in parallel at very low cost. Using these two terms, we calculate the regularizer given in Equation 3.

### 4.1 QUADRATIC LOSS

We begin with a linear regression toy problem where each input $x \in \mathbb{R}^d$ (with $d = 12000$) is sampled from a Gaussian distribution whose covariance follows a power-law spectrum ($\lambda_i \propto 1/i^{1.1}$ for $i = 1, ..., d$) that mimics the spectrum for Hessians observed in modern neural networks. The target is given by $w^{*\top} x$ for a predetermined $w^*$.

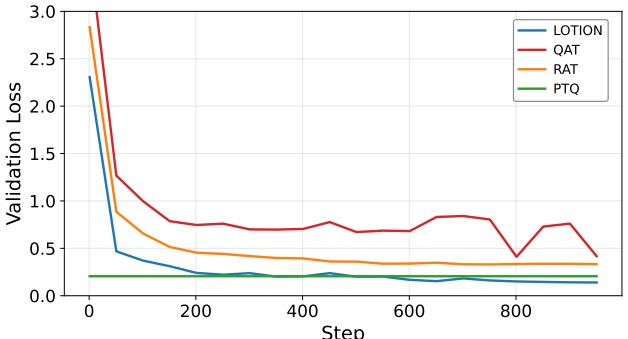

Figure 2: A comparison of INT4 quantized/rounded validation loss between LOTION, QAT, and PTQ, with summary table. We quantize using round-to-nearest (RTN) and randomized rounding (RR) and report the variant that yields the best performance. Full results are shown in Figure 7.

We train with SGD and compare quantized validation loss across all methods by directly quantizing weights at checkpoints throughout training via round-to-nearest (RTN) or randomized rounding (RR). As shown in Figure 2, LOTION outperforms both the PTQ and QAT methods in quantized validation loss. The behavior of QAT on the quantized validation loss is jagged, and runs optimized using this method quickly plateau, while LOTION continues to decrease the quantized validation loss. Our PTQ baselines are obtained by quantizing the target $w*$ via RTN/RR. Despite this, our LOTION runs outperform this PTQ baseline, suggesting that we are finding a full precision point that yields better quantized performance.

## 4.2 LINEAR NETWORK

In this section, we use our toy model to explore whether scaling the model can compensate for quantization noise, thereby eliminating meaningful performance differences between methods. We conduct experiments on a two-layer linear network given by

$$f(x) = \frac{1}{k} W_2 W_1 x$$

where $W_2 \in \mathbb{R}^{1 \times k}, W_1 \in \mathbb{R}^{k \times d}$ and $x \in \mathbb{R}^d$. We again consider the input distribution to be Gaussian with $d = 12000$ with a power-law decaying spectrum given by $\lambda_i \propto \frac{1}{i^{1.1}}$. The targets are given by $y = w^{*\top}x$, for $w^*$ sampled from a Gaussian distribution. We train with gradient descent, using the exact population hessian. The following lemma holds for the above network as $k \to \infty$.

**Lemma 4.** *For the uniform INT fine-grained, shared-scale quantization scheme, the quantized loss for $f(x)$ goes to 0 as $k \to \infty$.*

The above lemma holds as all the elements of the outputs $W_2$ can be set to 1 and each row of the first layer $W_1$ can be equal to a randomly rounded $w^*$.

We expect that as $k$ grows, the smoothed loss and the quantized loss for various methods will decrease. For each value of $k$, we plot the lowest quantized loss achieved at this model size. PTQ models are trained in full precision and randomly rounded or rounded-to-nearest to measure quantized loss. Our Ground-Truth (GT) baselines are initialized from models where all elements of $W_2$ are set to 1 and where the rows of $W_1$ are w*. These models are then randomly rounded or rounded-to-nearest. Following Lemma 4, GT's rounded loss goes to 0 as $k \to \infty$. As shown in Figure 3, LOTION outperforms all methods in quantized loss even as we scale up the model size.

## 4.3 LARGE LANGUAGE MODELS

In addition to our experiments with the synthetic setting, we pre-train models at 150M- (Figure 9) and 300M- parameter scales (Figure 4) to assess whether LOTION outperforms traditional QAT methods at realistic model sizes. We train and evaluate our models on C4 (Raffel et al., 2020) using OLMo (Groeneveld et al., 2024) following the hyperparameter settings of (Zhao et al., 2025). Both

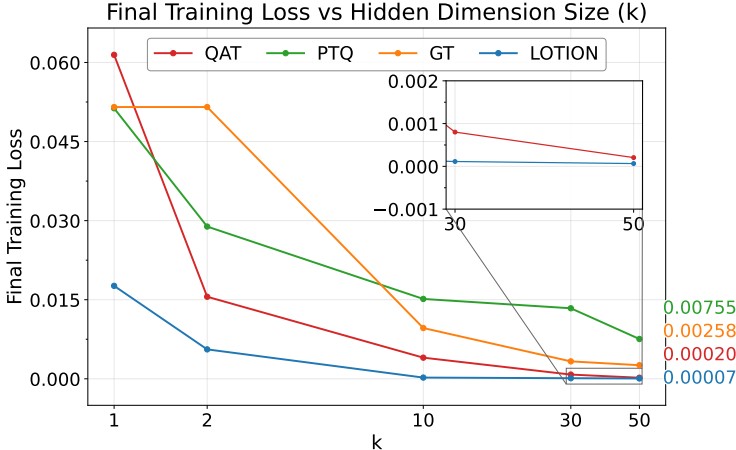

Figure 3: Final quantized training loss as a function of the hidden dimension, $k$, of a two layer linear network for LOTION, QAT, GT, and PTQ. Full results are shown in Figure 8.

sets of models are trained with Chinchilla-optimal token budgets (20x as many tokens as model parameters) (Hoffmann et al., 2022). Unlike in the synthetic experiments above, calculating the full Hessian is unrealistic for large language models. As described in Section 3, we use the empirical Fisher approximation as we would with Adam. Additionally, we do not differentiate through the empirical Fisher for computational efficiency. Finally, given that we are using the empirical Fisher as an approximation, in practice, we weight the regularization term by a hyperparameter, $\lambda$. This approach is agnostic to both the quantization format and the optimizer used, so we can easily adapt it to new quantization formats.

### 4.3.1 150M MODELS

As shown in Figure 9, even when training large language models, LOTION outperforms both QAT and PTQ baselines on quantized loss, especially at lower bit-width. At INT4 precision, both the round-to-nearest and randomly rounded validation loss continue to decrease when optimizing with LOTION faster than with QAT. PTQ performance seems to stall altogether. To validate whether this performance gap between QAT and LOTION persists as we continue to train, we reproduce the experiment with a larger training data budget (5x Chinchilla or 100x as many tokens as model parameters) and plot these results in Figure 1. LOTION continues to decrease the quantized/rounded validation loss while QAT's performance plateaus.

### 4.3.2 300M MODELS

As in the linear regression toy setting, we explore whether scaling the model size can compensate for quantization noise. We rerun our sweeps from Section 4.3.1 with a 300M parameter model and plot our results in Figure 4. As with the 150M models, LOTION outperforms QAT and PTQ at both INT4 and INT8 precision.

### 4.3.3 FP4 QUANTIZATION

Quantization to FP4 is generally favored over INT4 because of its non-uniform quantization scheme that can represent small values while accounting for rare large outliers. This increased resolution for values near zero tends to lead to lower overall quantization error and higher inference accuracy (Mehta et al., 2025). As the quantization error drops, the performance gaps between methods may become negligible.

However, in Figure 5, we validate that this is not the case and that LOTION grants performance gains over QAT even with this modern precision format. Although PTQ performance is more stable compared to the integer quantization sweeps, its quantization validation loss plateaus much earlier than both QAT and LOTION.

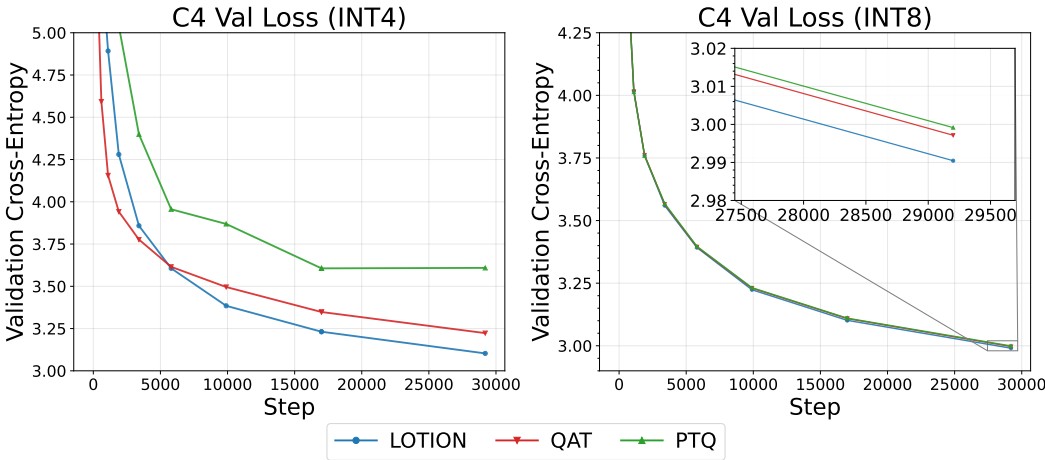

Figure 4: Quantized validation loss at INT4 (**Left**) and INT8 (**Right**) precision for LOTION, QAT, and PTQ on a 300M-parameter model. Full results are shown in Figure 11.

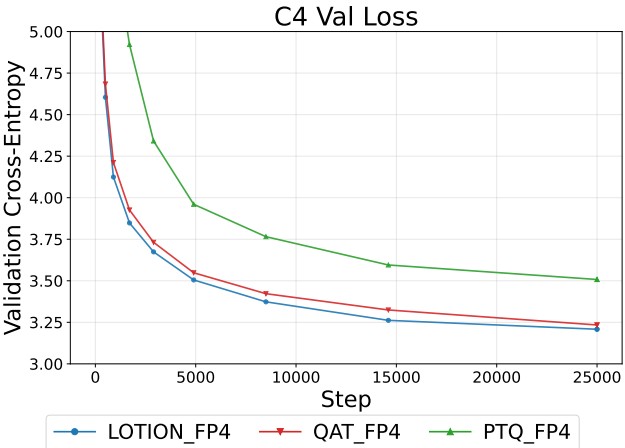

Figure 5: Quantized validation loss at FP4 precision for LOTION, QAT, and PTQ. Full results are shown in Figure 12.

## 5 DISCUSSION AND LIMITATIONS

With the increasing size of modern neural networks, low-precision execution is becoming a necessity for deployment. Thus, devising better mechanisms for obtaining quantization-friendly networks is an important challenge. Previous approaches like QAT use straight-through estimators but do not provide guarantees of convergence.

In comparison, LOTION smooths the loss surface of quantized loss while maintaining the global minima. This preserves the guarantees from the traditional optimization literature about convergence to a stationary point. Specifically, LOTION uses randomized rounding noise to transform the discontinuous loss surface into a continuous one that is differentiable almost everywhere.

However, when using randomized rounding, the loss surface is still not completely smooth due to the undefined derivatives at the quantization bin boundaries. Using other noise distributions for obtaining a smooth loss surface while preserving the global minima property is an interesting research direction. Finally, another promising direction for future research is extending LOTION to activation quantization to further improve compute and memory efficiency.

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

# A    APPENDIX

## A.1    VISUALIZATION OF LOTION

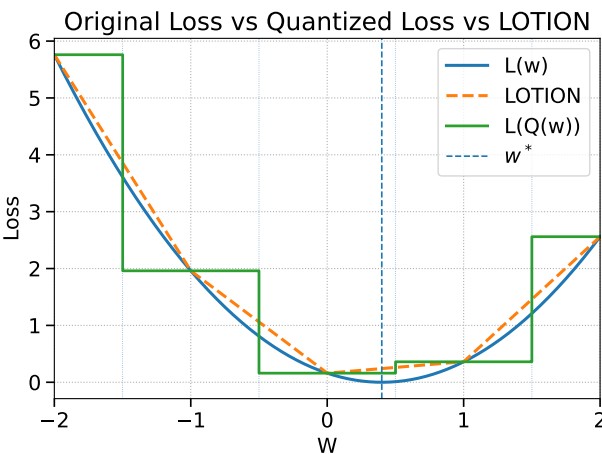

Figure 6: A simplified depiction of LOTION with a quadratic loss. We compare LOTION with the original loss $L(w)$ and the quantized loss $L(Q(w))$. The quantized loss surface is both non-smooth and discontinuous. LOTION yields a continous loss that shares the same minima as the quantized loss.

## A.2    ADDITIONAL DETAILS ON RANDOMIZED ROUNDING

### A.2.1    PROOF FOR LEMMA 1

*Proof.* Consider any $w_0 \in \mathbb{R}^d$. As $f$ is continuous, it implies that for any $\varepsilon > 0$, $\exists \delta$, such that $\|w - w_0\| \leq \delta \implies \|f(w) - f(w_0)\|_{W_2} \leq \varepsilon$.

$W_2$ distance being less than $\varepsilon$ indicates that there is a joint distribution $\gamma(x, y)$ with $x \in \mathrm{spt}(f(w))$, $y \in \mathrm{spt}(f(w_0))$ such that $\int \|x - y\|^2 \gamma(x, y) \leq \varepsilon^2$, with $\gamma(x, y)$ satisfying $\int_x \gamma(x, y) = f(w)$ and $\int_y \gamma(x, y) = f(w_0)$. Now, we can see that

$$\left| \mathbb{E}_{q \sim f(w)}[L(q)] - \mathbb{E}_{q' \sim f(w_0)}[L(q')] \right| = \left| \int [L(x) - L(y)] \gamma(x, y) \right|$$

$$\leq \int |L(x) - L(y)| \, \gamma(x, y)$$

As $f$ is locally bounded and $L$ is continuous, therefore, for any $x \in \mathrm{spt}(f(w))$ and $y \in \mathrm{spt}(f(w_0))$, $|L(x) - L(y)| \leq \zeta \|x - y\|$. Thus, we get

$$\left| \mathbb{E}_{q \sim f(w)}[L(q)] - \mathbb{E}_{q' \sim f(w_0)}[L(q')] \right| \leq \zeta \int \|x - y\| \gamma(x, y)$$

$$\leq \zeta \left[ \int \|x - y\|^2 \gamma(x, y) \right]^{1/2}$$

$$\leq \zeta \varepsilon$$

Thus $\mathbb{E}_{q \sim f(w)}[L(q)]$ is also continuous.    $\square$

### A.2.2    PROOF FOR LEMMA 2

*Proof.* For any $w \in \mathbb{R}^d$, $\mathrm{spt}(f(w)) \subseteq Q$. This implies that $\mathbb{E}_{q \sim f(w)}[L(q)] \geq \min_{w \in \mathbb{R}^d} L(cast(w))$.

However, for any $w \in Q$, $\mathrm{spt}(f(w)) = \{w\}$. This implies that $\min_{w \in \mathbb{R}^d} \mathbb{E}_{q \sim f(w)}[L(q)] \leq \min_{w \in \mathbb{R}^d} L(cast(w))$.

Combining the two equations above, we can say

$$\min_{w \in \mathbb{R}^d} \mathbb{E}_{q \sim f(w)}[L(q)] = \min_{w \in \mathbb{R}^d} L(cast(w))$$

$\square$

### A.2.3 PROOF FOR LEMMA 3

*Proof.*

$$\mathbb{E}_\varepsilon[\nabla\mathcal{L}(w + \varepsilon)] = \mathbb{E}_\varepsilon[H(w + \varepsilon - w^*)]$$
$$= H(w - w^*)$$

because $E[\varepsilon] = 0$. $\square$

### A.2.4 EXAMPLE: SHARED-SCALE INTEGER ROUNDING

We provide an example of a randomized rounding scheme corresponding to the casting function defined in Section 2.1. Consider a scalar $z_i'$ as defined below:

$$z_i' = \frac{w_i}{s_B}.$$

The randomized rounding scheme is defined as below:

$$\mathrm{RR}(w) = \begin{cases} s_B z_i' & \text{if } z_i = z_i' \\ s_B \lfloor z_i' \rfloor & \text{w.p. } \lceil z_i' \rceil - z_i' \\ s_B \lceil z_i' \rceil & \text{w.p. } z_i' - \lfloor z_i' \rfloor \end{cases}$$

We use this rounding scheme for LOTION with integer formats.

### A.3 ADDITIONAL RESULTS FOR SYNTHETIC EXPERIMENTS

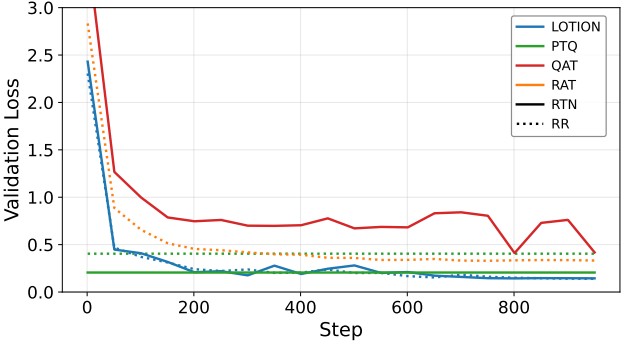

| Method | Val. loss |
|---|---|
| LOTION (RR) | 0.13988 |
| LOTION (RTN) | 0.14419 |
| PTQ (RTN) | 0.20566 |
| RAT (RR) | 0.33230 |
| PTQ (RR) | 0.40449 |
| QAT (RTN) | 0.79181 |

Figure 7: A comparison of INT4 quantized validation loss between LOTION, QAT, and PTQ, with summary table. We include the final validation loss after quantizing using round-to-nearest and randomized rounding.

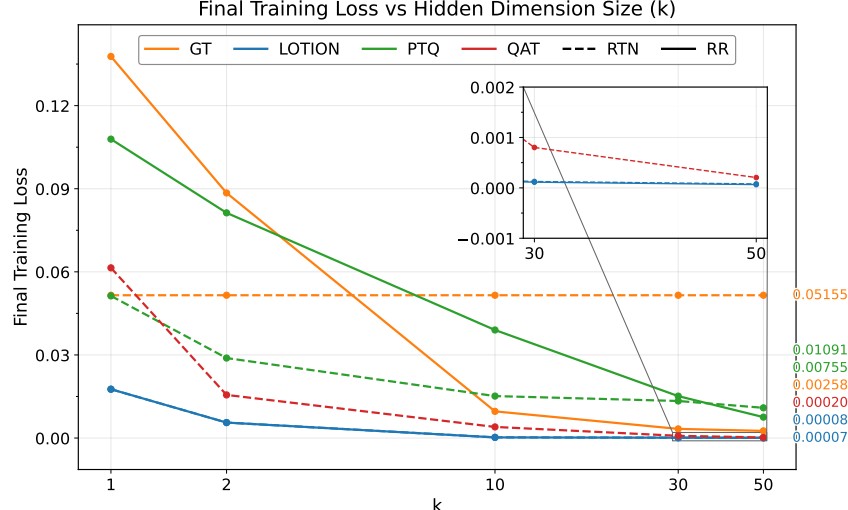

Figure 8: Final quantized training loss as a function of the hidden dimension, $k$, of a two layer linear network for LOTION, QAT, GT, and PTQ. Models are quantized to INT4 using round-to-nearest or randomized-rounding.

## A.4 ADDITIONAL RESULTS FOR LARGE LANGUAGE MODEL EXPERIMENTS

### A.4.1 150M-PARAMETER MODEL

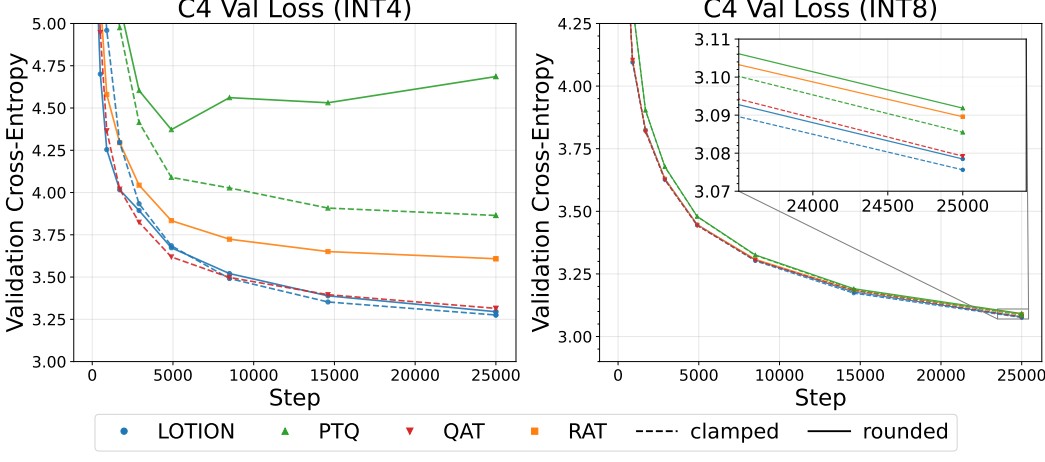

Figure 9: 150M model: Quantized and rounded validation loss at INT4 (**Left**) and INT8 (**Right**) precision for LOTION, QAT, RAT and PTQ. Table 1 contains the final quantized validation loss for each method.

Table 1: Final validation cross-entropy (150M)

| Method | Metric | INT4 | INT8 |
|---|---|---|---|
| PTQ | RR | 4.686 | 3.092 |
| PTQ | RTN | 3.864 | 3.085 |
| RAT | RR | 3.608 | 3.090 |
| QAT | RTN | 3.315 | 3.079 |
| LOTION | RR | 3.295 | 3.078 |
| LOTION | RTN | **3.276** | **3.076** |

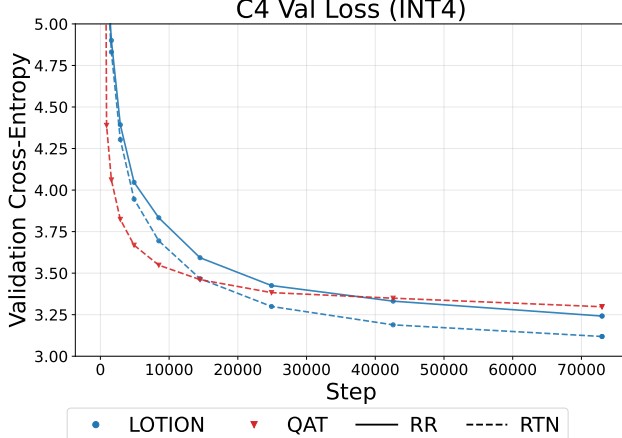

Figure 10: Round-to-nearest and randomized rounding validation loss at INT4 precision for LOTION and QAT on a 150M-parameter model trained to 5x chinchilla (100x as many tokens as parameters).

## A.4.2 300M-PARAMETER MODEL

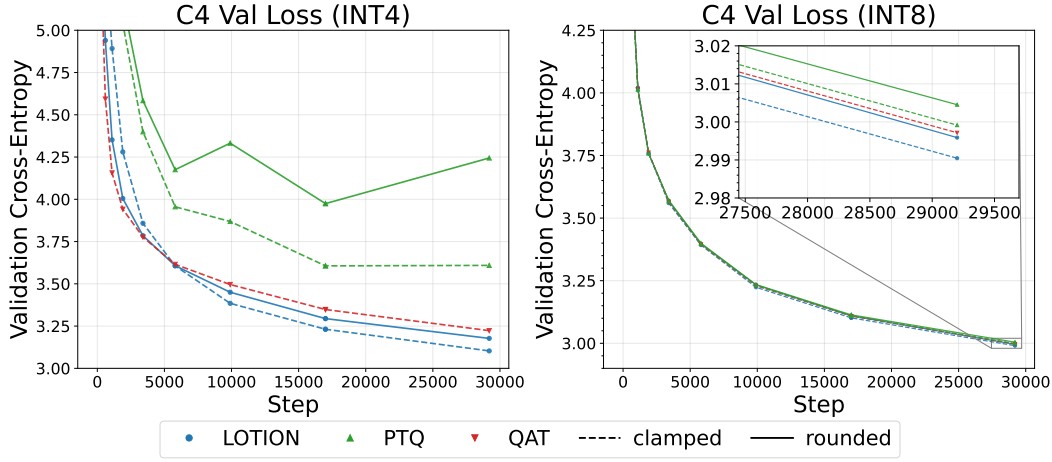

Figure 11: 300M model: Quantized validation loss at INT4 (**Left**) and INT8 (**Right**) precision for LOTION, QAT, and PTQ. Table 2 contains the final quantized validation loss for each method.

Table 2: Final validation cross-entropy (300M)

| Method | Metric | INT4 | INT8 |
|--------|--------|------|------|
| PTQ | RR | 3.9745 | 3.0045 |
| PTQ | RTN | 3.6062 | 2.9992 |
| QAT | RTN | 3.2230 | 2.9972 |
| LOTION | RR | 3.1772 | 2.9959 |
| LOTION | RTN | **3.1031** | **2.9905** |

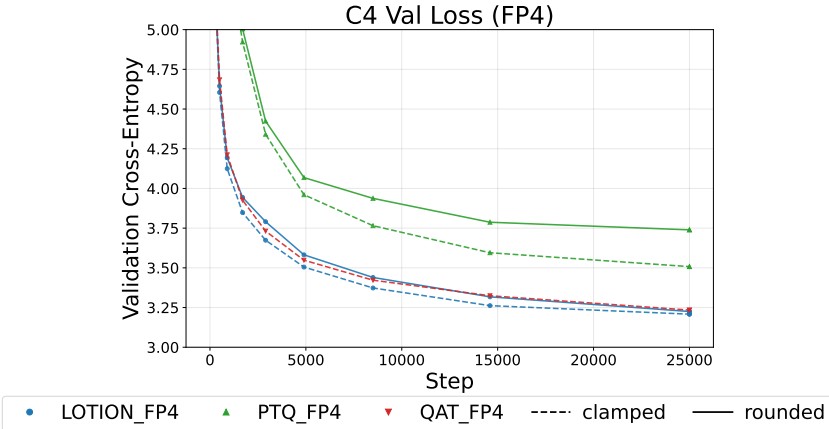

Figure 12: Quantized and rounded validation loss at FP4 precision for LOTION, QAT, and PTQ.

### A.4.3 FP4 QUANTIZATION

## A.5 HYPERPARAMETERS

### A.5.1 LINEAR REGRESSION

| Shared Hyperparameters | |
|---|---|
| Learning Rate | 3.0e-6, 3.0e-5, 3.0e-4, 3.0e-3, 1.0e-2, 3.0e-2, 1.0e-1, 3.0e-1, 6.0e-1, 8.0e-1 |
| LR Scheduler | Cosine |

### A.5.2 TWO LAYER NETWORK

| Shared Hyperparameters | |
|---|---|
| Learning Rate | 0.0003 0.003 0.03 0.1 0.3 0.6 1.2 |
| LR Scheduler | Cosine |

### A.5.3 LANGUAGE MODEL

| Shared Hyperparameters | |
|---|---|
| Learning Rate | 3.16e-4, 1.0e-4, 3.16e-3, 1.0e-3, 3.16e-2, 1.0e-2 |
| Weight Decay | 0 |
| **LOTION specific** | |
| Lambda (regularization coefficient) | 3000, 10000, 30000, 100000 |

