# OpenReview forum: "LOTION: Smoothing the Optimization Landscape for Quantized Training"
_ICLR.cc/2026/Conference — Submitted to ICLR 2026_

### Official Review · Reviewer_vART · 2025-10-30

**Soundness:** 3
**Presentation:** 2
**Contribution:** 2
**Rating:** 4
**Confidence:** 3

**Summary:**

This paper proposes a novel quantized training smoothing framework called LOTION, which introduces randomized rounding noise to smooth the quantized loss, addressing the problem of gradient blockage and optimization instability caused by the discontinuity of the loss function in traditional quantization training.

**Strengths:**

This paper introduces a theoretically grounded approach to quantized training that smooths the quantized loss via randomized noise, effectively mitigating gradient instability and convergence issues in traditional QAT while maintaining conceptual simplicity and clear theoretical interpretability.

**Weaknesses:**

Although the paper proposes a reasonable smoothing framework from a theoretical perspective, it lacks an in-depth analysis of gradient stability near quantization boundaries. The transitions between sections are somewhat abrupt, the figures are not highly distinctive, and the experimental results focus mainly on loss values without providing richer performance comparisons or ablation studies.

**Questions:**

1、	Although the paper mentions that absolute maximum quantization can prevent overflow issues, LOTION requires computing or approximating the diagonal entries of the Hessian (e.g., via the empirical Fisher) at each training step. When the model scales to billions of parameters (such as the 150M/300M models mentioned in the paper), is this approach still feasible? Does it lead to lower efficiency compared with other methods?
2、	It is recommended that the authors provide more experimental details and include additional comparative experiments, rather than only comparing loss values.
3、	It is recommended to include an algorithmic flowchart to make the proposed method clearer.
4、	The authors mention that LOTION constructs a continuously differentiable loss function through expectation smoothing, but its gradients exist only “almost everywhere.” Have the authors examined the stability of these gradients near quantization boundaries? Could gradient explosion or vanishing occur in such regions?
5、	The overall writing is fluent, but the authors could reduce repetitive expressions (such as “in particular” and “specifically”) to improve readability.

---

> ### Author Response · Authors · 2025-11-20
>
> We thank the reviewer for their positive assessment of LOTION’s principled formulation and motivation and for the constructive suggestions. We address these concerns below:
>
> **Feasibility at scale**
>
>
> 1. LOTION is still feasible at scale. There is minimal compute/memory overhead. In our implementation, we use the per batch empirical Fisher which has the same cost and footprint as Adam’s second moment (if reusing this, then there is no additional param state at all). The only additional computation is the tiny overhead of backpropagating through the error-variance term. These computations are also done on the fly, so, aside from the per-block scale, there are no extra activations/tensors being stored.
>
>
> **Additional comparative experiments**
>
>
> 1. What comparative experiments beyond comparing loss values would the reviewer like to see? Additionally, what experimental details does the reviewer think are lacking beyond what is provided in section 4.3?
>
>
> **Algorithmic flowchart**
>
>
> 1. We will make the actual algorithm clearer. Could the reviewer explain what kind of algorithmic flowchart they would like to see?
>
>
> **Gradient stability near quantization boundaries**
>
>
> 1. Gradients exist near quantization boundaries, and we empirically demonstrate training stability across a wide range of experiments.
>
>
> **Writing Clarity**
>
>
> 1. Thank you for your suggestions. We will improve the writing clarity.
>
> If our response has resolved the concerns of the reviewer, we would be grateful if they can update the score accordingly.

---

### Official Review · Reviewer_YHM6 · 2025-10-31

**Soundness:** 3
**Presentation:** 3
**Contribution:** 2
**Rating:** 4
**Confidence:** 3

**Summary:**

The paper proposes a principled framework, called LOTION, for training quantized neural networks by smoothing the discontinuous loss surface caused by quantization. Instead of relying on heuristic Straight-Through Estimators (STE), LOTION computes the expected quantized loss under randomized-rounding noise, which may make the loss continuous and differentiable almost everywhere while preserving the original global minima. The authors derives a closed-form second-order approximation showing the method acts as a curvature-aware regularizer, stabilizing quantized training. The paper also demonstrates empirically that LOTION outperforms standard QAT and PTQ on both synthetic tasks and large-scale 150M–300M parameter language models at INT4, INT8, and FP4 precision.

**Strengths:**

1. The paper proposes a new and principled formulation of quantized optimization through loss-level smoothing. It is novel by bridging classical randomized smoothing theory with quantization-aware training.
2. The theoretical part seems solid with clear derivations and a well-justified second-order analysis linking smoothing to curvature-aware regularization.
3. The generality and simplicity of LOTION may have adoption potential across both research and downstream applications.

**Weaknesses:**

1. While results on 150M–300M language models convincingly support the main claims, the experiments are limited to small parameter sizes and transformer-based architectures. The method’s generality would be more compelling if evaluated on other architectures/models.
2. Although the paper emphasizes that LOTION requires no new hyperparameters, it would be better to have a straightforward training-time cost or memory overhead compared to QAT.
3. Lack of Ablation on Smoothing Strength $\lambda$ and Noise Distribution
4. The paper only reports val loss but not downstream metrics (e.g., perplexity, BLEU)
5. The paper proves that the smoothed loss preserves the global minima of the quantized objective, but it lacks a bound on how closely the smoothed loss approximates the original during optimization

**Questions:**

1. The authors mentioned that extending LOTION to activation quantization as future work. Then would randomized rounding for activations require dynamic smoothing (per-batch or per-layer), and would this break the differentiability guarantees?
2, Did anuthors observe any stability issues or over-smoothing when $\lambda$ is too high? How sensitive are the results to $\lambda$?

---

> ### Author Response · Authors · 2025-11-20
>
> Thank you for highlighting LOTION’s principled formulation, adoption potential, and the theoretical section of our draft. We address your concerns below:
>
> **Other architectures/model-sizes**
>
>
> 1. Although we believe that LOTION is applicable to other architectures, our goal is to validate the stability of optimizing the quantized loss at low-bit widths on language modelling, so we focus on transformers. Most recent quantization literature has been centered on LLMs as well. We are happy to revise the abstract to clarify/emphasize this point
> 2. We chose these model sizes because of computational concerns. We can run 600m experiments and would be happy to provide the results if you would like
>
>
> **Overhead concerns**
>
>
> 1. There is minimal training-time cost and memory overhead for LOTION. We use the per batch empirical Fisher which has the same cost and footprint as Adam’s second moment (if reusing this, then there is no additional param state at all). The only additional computation is the tiny overhead of backpropagating through the error-variance term. These computations are also done on the fly, so, aside from the per-block scale, there are no extra activations/tensors being stored. (In our experiments, QAT is actually slower because we need to store two copies of the weights, quantized and full precision, and switch between them throughout training)
>
>
> **Lack of ablation on lambda and noise distribution:**
>
>
> 1. For the synthetic experiments, we use lambda=0.5 because this value comes from the second order expansion of the smoothed loss. For the LLM experiments, we tune lambda (the values that we test are listed in Appendix A.5.3). We believe that we need to tune lambda because we are using the empirical fisher for our Hessian approximation. If using the actual Fisher, we believe the default lambda of 0.5 should work well. As mentioned in the conclusion, ablations over noise distributions will be included in future work. We are choosing stochastic rounding noise for now because it is zero mean and closely resembles the standard quantization operation.
>
>
> **Downstream metrics**
>
>
> 1. We reported validation loss trajectories because we are concerned with training stability with LOTION and optimization of the quantized objective. As shown in previous papers, perplexity correlates closely with downstream metrics. Additionally loss spikes, divergence, and time to convergence are visible in the loss curves even though they may not be reflected in the final downstream accuracy. We are limited in compute and time but are happy to run these downstream evals if they will change the reviewer’s score.
>
>
> **Bound on approximation to original quantized loss**
>
>
> 1. This is beyond the scope of the paper. We also do not want the loss surface to be similar to the original quantized loss as the original is not differentiable. We show that the global minima of this quantized loss surface are preserved
>
>
> **Questions**
>
>
> 1. With respect to the activation quantization, you are correct that the rounding would be done per batch, so the objective would be a function of both the weights and the batch of data. However, for a given batch, the function would still be differentiable
> 2. We do need to tune lambda, but we believe that this comes from the fact that we are using the empirical fisher. We believe that if using the actual fisher, our default lambda of 0.5 will work well.
>
> If our response has resolved the concerns of the reviewer, we would be grateful if they can update the score accordingly.

---

### Official Review · Reviewer_9zKi · 2025-11-01

**Soundness:** 4
**Presentation:** 3
**Contribution:** 3
**Rating:** 8
**Confidence:** 4

**Summary:**

This paper addresses the discontinuities introduced during quantization aware training (QAT) that relies on straight-through-estimators (STE) during backward pass. The authors formulate a smoothened loss function that alleviates the discontinuities introduced during QAT. This is achieved by formulating the loss objective as an expectation over randomized rounding operations. Theoretical analysis shows that doing this renders the smooth loss objective almost differentiable, and can be enforced as tractable regularizers. Experiments on synthetic tasks and LLMs shows improved convergence behaviour for the proposed optimization scheme (LOTION).

**Strengths:**

* Smoothing of quantized loss landscape by approximating with continuous loss landscape, and local minimum convergence guarantees.
* Improved performance on QAT on wide-range of models.
* Trains the model on the expectation of the quantized loss under randomized-rounding noise
* Theoretical analysis of starting with quadratic losses to arrive at the implied regularization, and then extending it to the general case is convincing.
* Compelling experiments on synthetic tasks and on LLMs show that the proposed smoothing of loss function helps consistently.

**Weaknesses:**

* **Randomized rounding versus stochastic rounding:** The authors use stochastic rounding (the more established term in literature) to describe related work, but in their own method description they adhere to randomized rounding. Is this for a specific reason? To the extent I see, they are exactly the same. Or is there a subtlety that the authors want to point out, and clarify?

* **Beyond block-wise quantization:** The formulation of the method is presented for block-wise quantization. Is there a specific reason to limit to these settings?
* What do the authors mean by this statement:
  > The scale parameters {sB } depend on wi , so the quantization lattice moves as optimization proceeds.
* **Data-dependent regularizer:** In L239, authors mention "a data-dependent" regularizer; I am unsure how the claim of data-dependence is manifested here?
 * $L_{GN}(w)$, in its final formulation in Eq. 3 is interpreted as a curvature-aware ridge regularizer. And the authors state these components can be obtained "using another backpropagation with sampled labels", or "empiricial Fisher approximation". These are important choices that can have significant implications on the optimisation. Which is the specific choice used in the work; as far as I see, it is the Fisher approximation (L400)? What would be better, and in which settings?
* In Fig 3, why do the gains between QAT and LOTION diminish for more complex networks (as k increases).
* Why do the differences in performance diminish between INT4 and INT8 levels? Is it because the quantized loss landscape more drastic/discontinuous in INT4 regime?
### Other comments

* LLMs not abbreviated in first sentence in Introduction.
* "underperforms QAT" -> "underperforms compared to QAT"
* $H$ is not defined in its first use in L214.
* Reporting the wall-clock time for convergence when compared to QAT will be interesting to see.

**Questions:**

See points above in weaknesses.

---

> ### Comment · Reviewer_hfWb · 2025-11-15
>
> Dear Reviewer 9zKi,
>
> Although it is not my responsibility, let me explain some concerns you raised about the submission.
>
> By "The scale parameters {sB } depend on wi , so the quantization lattice moves as optimization proceeds.", I think the authors meant that during training, w_i's are changing, so s_B and the integer quantized values (the so-called quantization lattice) are both changing during the training/optimization process.
>
> For "Data-dependent regularizer", the Hessian and also s_B's and Delta's are obtained during training and optimized with the training data, so they are data-dependent.
>
> Regarding the Hessian for the loss, the authors seems to use the exact one for the toy models of one or two linear layers, as in Line 359, and for the real data with larger models, they use the estimator you mentioned to avoid computational complexity. The question you asked about the comparison is interesting.

---

> > ### Author Response · Authors · 2025-11-20
> >
> > We are sincerely grateful for your thoughtful assessment and generous summary of our contributions. We especially appreciate the clear articulation of our goal and that you found our theoretical analysis from quadratic losses to general LLM case convincing. Thank you also for your helpful comments. We address each of them below:
> >
> >
> > **Randomized rounding versus stochastic rounding (notation)**
> >
> >
> > 1. Thank you for pointing this out. You are absolutely right that we do use standard stochastic rounding, and we will change this in the draft to avoid confusion.
> >
> >
> > **Beyond block-wise quantization**
> >
> >
> > 1. Great question. We present LOTION for block-wise quantization to reflect new precision formats like NVFP4 and MXFP4 ((https://developer.nvidia.com/blog/introducing-nvfp4-for-efficient-and-accurate-low-precision-inference/ ; https://arxiv.org/html/2412.19437v1#S3) and other per-channel/per-group scaled formats. Our goal was to demonstrate how LOTION interacts with shared-scale effects, but the objective itself is not limited to block-wise quantization formats: LOTION can be used for other formats by changing the scale granularity.
> >
> >
> > **Moving Lattice**
> >
> >
> > 1. As Reviewer hfWb mentioned, in these shared scale formats, the scale is a function of the (abs-max value of) the weights, so as w updates throughout training, the exact set of quantized points (the quantization lattice) will change.
> >
> >
> > **Data-dependent regularizer + Hessian approximation**
> >
> >
> > 1. The “hessian” term in our regularizer depends on the data. As you pointed out, we use the empirical Fisher approximation for computational overhead reasons. We already have access to this because we must compute gradients for the optimizer anyway. We believe that the extra backprop would help but will add wall-clock time and memory overhead
> >
> >
> > **Toy model - diminishing gains**
> >
> >
> > 1. Great observation. We believe that the increased capacity compensates for the quantization noise. The problem is getting easier as we increase the model size, so the set of possible solutions grows with k, making it easier for qat to find a good solution.
> >
> >
> > **INT4 vs INT8**
> >
> >
> > 1. INT4 is indeed a harder quantization problem: The fewer bits you have per value, the more quantization noise there will be. In this case where the noise is greater, we expect an approximation like STE/QAT to perform much worse.
> >
> >
> > **Other Comments**
> >
> >
> > 1. Thank you very much for reading our paper so carefully. We will incorporate your edits in the into the final draft

---

### Official Review · Reviewer_hfWb · 2025-11-01

**Soundness:** 1
**Presentation:** 1
**Contribution:** 1
**Rating:** 0
**Confidence:** 4

**Summary:**

This paper proposed a principled smoothing framework with randomized rounding for quantization aware training. The authors conducted  experiments for INT4, INT8, and FP4 quantization for 150M and 300M parameter language models.

**Strengths:**

It is beyond my capability to find strength in the paper.

**Weaknesses:**

The writing can be improved.

**Questions:**

I do not know what questions to ask.

---

> ### Comment · Reviewer_hfWb · 2025-11-15
> **I would like to revise my review and apologize to the authors for previous misunderstanding**
>
> Dear Authors,
>
> Thanks for the help and the concerns pointed out by other reviewers and ACs. I re-read the paper and found that my previous review is not appropriate. I would like to apologize for it and would like to change my rate. I am also OK if my review is discard.
>
> For strengths, the paper propose a method named LOTION (from my understanding, there is no formal definition of it except a related concept at the beginning of Section 3, but the formula seems to be in Eq. 2 and 3 in the original submission) based on randomized rounding (Definition 1 of Section 3.1 in the original submission) and rounding-aware training (defined at Line 245 in the original submission). This method, to my knowledge, is new and interesting. The authors mainly focused on the introduction, definition, and formal verification with toy examples and theorem proof to demonstrate their method, and finally verify their method via pretraining two models of size 150M and 300M on real world dataset. Through experiments and verifying on Validation loss, they demonstrate that the loss can be optimized better to lower values continuously, and outperforms over conventional methods of QAT and PTQ, especially for low precisions like INT4 (Fig. 4) and FP4 (Fig. 5) on real data.
>
> **Strengths:**
> 1. The authors provide a novel principled quantization method based on randomized rounding and analysis of the statistics of loss, to introduce an extra quadratic term (Eq. 1 and 2 in the original submission).
> 2. Specifically, to make the loss properly behaved (i.e., avoiding negative curvature), the authors introduce Gauss-Newton replacement for the extra quadratic term in their loss, due to the positive-semidefinite property of Gauss-Newton matrix.
> 3. The authors verified the effectiveness and scaling-up behavior of their method on toy examples (Fig. 3 in the original submission), and verified their method on real small models on small real dataset (Fig. 1, 4, and 5 in the original submission).
>
> **Weaknesses:**
> 1. The writing can be improved. For example, it is not easy to follow where LOTION is defined. In Section 3 in the original paper, the authors first introduce the general concept of loss smoothing, and the randomized rounding (RR) and round-aware training (RAT), and afterwards seems to introduce the formula for LOTION. A better way is to define clearly what is LOTION after the discussion on the RAT.
> Another case is at line 270 in the original submission, where the author introduce the Gauss-Newton replacement. A brief introducing sentence here can be better. For example, "To leverage the Hessian introduced above for our case of neural network f, where the loss is l defined earlier in Equation ... (Line 142 in the original submission), here we explicitly give the Hessian and analyze the positiveness of different components. ..."
> Also, each equation, if occupying a single line, should be numbered or labeled, even if it was not the main equations or not referenced. Examples violating this rule include Lines 142, 146, 161, 176, 203, 214, 226, 230, 237, 248, 258, 264, 268, 273, 284, 355 in the original submission.
> 2. There are some points that symbols or notations are introduced without clear description, making it more difficult to understand the meaning of them or the sentence, or even the whole lemma or theorem. Examples include the following:
>
>       a. In Definition 1, what does P[Q] mean? What is W2 distance on P[Q]?
>
>       b. In Lemma 1, it could be better to say "... continuous w.r.t. w in terms of L2 norm ..." instead of "... is continuous w.r.t. L2 norm ...".
> 3. The authors mentioned "the spectrum for Hessians observed in modern neural networks" at Line 323 and Line 357 in the original submission. Could the authors provide reference for this?
> 4. There are some typos or miswriting. For example, at Line 179 in the original submission, there should be no "then" before the loss expression. In the equation at Line 226 in the original submission, the "w.p." seems to be "with probability", but a little difficult to understand from the first reading.
> 5. The experiments on INT8 (Fig. 4 in the original submission) and FP4 (Fig. 5 in the original submission) show that the proposed method might not be very useful for higher precision quantized models. If the authors could provide some results on lower precision like INT2, it might be better (but it's OK if such experiments are not easy to finish during the rebuttal period).
>
> [to be continued]

---

> ### Comment · Reviewer_hfWb · 2025-11-15
> **[Continued] I would like to revise my review and apologize to the authors for previous misunderstanding**
>
> **Questions:**
> 1. In part 3 of Definition 1 in the original submission, what does it mean by "RR(w) has probability 1 on w"?
> 2. In Figure 2, the authors compare LOTION using RR (random rounding) with PTQ and QAT using RTN (rounding to nearest), and RAT using RR. It seems that LOTION incorporates two parts, one is adding the proposed extra Hessian term to the loss, and the other is RR. Could the authors also provide results for adding the Hessian loss with rounding to nearest? After all, if the weights are distributed almost random (and this is usually the case for typical neural networks where after training the weights are still apparently Gaussian distributed), the expectation should behave similar to random rounding. With this experiments, it can help to analyze whether the benefits mainly come from the extra loss term, or the random rounding is also crucial.
> 3. In Line 369, where the authors mentioned the experiments with Ground-Truth (GT) baselines, the second layer is initialized with all set to 1. Although this is not a deep network and thus not suffer much from gradient exploding or vanishing, such initialization scale still might be suboptimal, especially if the weight scales from different layers are not homogeneous and learning rates for them are not properly set. Could the author provide an improved GT experiments, where the second layer is still initialized to be the same for each component (i.e., proportional to the all-one vector), but with some proper scale of, say Xavier or Kaiming initialization, so the gradients for the weights are properly scaled, weight quantization is still possible, and the requirement in Lemma 4 can still be guaranteed.
> 4. The proposed method seems to introduce two hyperparameters, i.e., the block size B for quantization, introduced in Line 158 in the original submission, and the loss strength term $\lambda$ introduced for real model in Line 402 in the original submission. How does these two hyperparameters impact the final results, and how are they determined in practice? Could the authors provide more ablation study on them?
> 5. For results presentation, why the authors mostly only show the results on Validation loss curve? Could the author provide a table on final metrics like accuracy or something else to compare those different methods listed in the paper on the real dataset?
>
> Based on the above revised review, I would like to change my current rate accordingly. If the authors could respond to my revised review properly, I would like to consider raise my score.
>
> I would like to apologize again for my previous review. Hope the revised one is more appropriate.
>
> Best,
> Reviewer hfWb

---

> > ### Author Response · Authors · 2025-11-20
> >
> > Thank you for re-reading the paper and for providing detailed comments and suggestions. We appreciate that you found LOTION to be a novel, principled quantization method that bypasses the issue of non-differentiability in typical quantization aware training. We also appreciate that you found the toy examples and LLM experiments compelling. Thank you also for your comments. We address them below and list any revisions we will make.
> >
> >
> > **Writing Clarity:**
> >
> >
> > 1. We agree that the definition of LOTION could be clearer. We will move the formal definition to the end of 3 after the loss smoothing as you mentioned and will add a boxed Algorithm to clearly define the algorithm. We will incorporate your suggestions on introducing the GN replacement and on numbering equations.
> >
> >
> > 2. P[Q] is the set of probability measures supported on the quantization lattice Q. W2 is the 2-Wasserstein distance on these probability measures. We agree that this is confusing and will add definitions to these and any other symbols/notations that are currently unclear. We will also change the phrasing for Lemma 1
> >
> >
> > 3. We will add these references supporting the empirically observed spectral properties of Hessians in modern neural nets:
> >
> >
> > 4. Thank you for spotting these typos. We will correct them.
> >
> >
> > **Questions about Rounding:**
> >
> >
> > 1. “RR(w) has probability 1 on w” in Definition 1.3 refers to the fact that if w = cast(w) (ie. w lies on a quantization bin boundary), then it is rounded to that point with probability 1.
> >
> >
> > 2. Although we would ideally directly optimize the smoothed loss, this is not possible in practice, so LOTION approximates the smoothed loss by taking a second-order expansion and adding one term that combines the curvature (hessian) with the error term from randomized rounding. We cannot provide results for the hessian term with RTN error because the error in this case is not differentiable.
> >
> >
> > 3. For your point about higher precision quantization, it is not that LOTION is not useful at higher precision, we expect it to still outperform QAT here, it is just that quantization loss will be lower with more bits, so the gaps in performance are not as pronounced. This gap in performance is particularly visible at extreme quantization.
> >
> >
> > 4. INT2: We do not have the compute to run these sweeps by the end of the rebuttal period but will look into this. We will finish it for the camera-ready draft
> >
> >
> > **Two layer linear net GT baselines**
> >
> >
> > 1. There seems to be a misunderstanding: these GT baselines are not trained. They are initialized to a ground truth solution (averaging copies of w* yields w*) and directly rounded or quantized. We believe that the “improved GT experiments” that you ask for are the PTQ baselines already provided in the experiment. What we are interested in demonstrating with the GT baselines is that 1. an optimal solution to the original problem may not be an optimal solution to the quantized loss and that 2. LOTION can outperform these baselines at different model sizes
> >
> >
> > **Hyperparameters**
> >
> >
> > 1. For the synthetic experiments, we use lambda=0.5 because this value comes from the second order expansion of the smoothed loss. However, as you note, we tune lambda for the LLM experiments (the values that we test are listed in Appendix A.5.3). We believe that we need to tune lambda because we are using the empirical fisher for our Hessian approximation. If using the actual Fisher, we believe the default lambda of 0.5 should work well. We do not introduce the block size parameter: it is inherent in many modern quantization formats like MX and NV formats (https://developer.nvidia.com/blog/introducing-nvfp4-for-efficient-and-accurate-low-precision-inference/ ; https://arxiv.org/html/2412.19437v1#S3).
> >
> >
> > **Metric Choice**
> >
> >
> > 1. We emphasized validation loss trajectories because we are concerned with training stability and optimization of the quantized objectives. Loss spikes, divergence, and time to convergence are visible in the loss curves even though they may not be reflected in the final downstream accuracy. We are limited in compute and time but are happy to run these downstream evals if they will change the reviewer’s score. Please let us know.
> >
> >
> > If our response has resolved the concerns of the reviewer, we would be grateful if they can update the score accordingly.

---

### Meta-Review · Area_Chair_9MVJ · 2026-01-02

**Summary:**

This paper proposed LOTION: Smoothing the Optimization Landscape for Quantized Training, which leverages empirical fisher information to improve QAT.

This paper received mixed review scores (0, 4, 4, 8). Reviewer hfWb’s original review is flagged as inappropriate by authors, reviewers and ACs. Reviewer hfWb substantially updated the review comments. I looked at the comments and discussion, while discarding the very low score 0.

Reviewers like the theoretical foundation of the paper, but also raised several concerns: the empirical experiments use only relatively small networks (150M & 300M transformers) and it is unclear if the method can scale; the comparison results use limited metrics (eval loss) and baselines (plain AQT, PTQ); the clarity of the method can be improved.

The authors responded to the above concerns during rebuttal. However, the short responses did not foster further engagement from reviewers with concerns (the two 4 reviews), and the concerns on scalability and comparison did not seem to be sufficiently addressed. I would encourage the authors to incorporate the reviewers’ feedback, and improve the draft on experiment scales, comparisons, and presentation clarity. For example, if the authors do not have enough computation resources, they may consider using parameter efficient training such as LoRA; quantization is extensively studied in

**Reviewer Concerns:**

Some of the clarification questions are answered, however, the short responses did not foster further engagement from reviewers . And the concerns on experiment scales and comparisons are not fully addressed.

**Reviewer Scores:**

Out of the (0, 4, 4, 8) scores, we can discard the extremely low score 0. The positive score 8 is confirmed. However, it is unclear whether the two 4 scores can be improved. Because of the discussion status, they might not be updated.

---

### Decision · Program_Chairs · 2026-01-26

Reject